# Synthetic Lethality Targeting Polθ

**DOI:** 10.3390/genes13061101

**Published:** 2022-06-20

**Authors:** Małgorzata Drzewiecka, Gabriela Barszczewska-Pietraszek, Piotr Czarny, Tomasz Skorski, Tomasz Śliwiński

**Affiliations:** 1Laboratory of Medical Genetics, Faculty of Biology and Environmental Protection, University of Lodz, 90-236 Lodz, Poland; mal.drzewiecka@gmail.com (M.D.); gabriela.barszczewska.pietraszek@edu.uni.lodz.pl (G.B.-P.); 2Department of Medical Biochemistry, Medical University of Lodz, 92-216 Lodz, Poland; piotr.czarny@umed.lodz.pl; 3Fels Cancer Institute for Personalized Medicine, Departament of Cancer and Cellular Biology, Lewis Katz School of Medicine, Temple University, Philadelphia, PA 19140, USA

**Keywords:** DNA damage, DNA repair, personalized medicine, polymerase theta, synthetic lethality

## Abstract

Research studies regarding synthetic lethality (SL) in human cells are primarily motivated by the potential of this phenomenon to be an effective, but at the same time, safe to the patient’s anti-cancer chemotherapy. Among the factors that are targets for the induction of the synthetic lethality effect, those involved in DNA repair seem to be the most relevant. Specifically, when mutation in one of the canonical DNA double-strand break (DSB) repair pathways occurs, which is a frequent event in cancer cells, the alternative pathways may be a promising target for the elimination of abnormal cells. Currently, inhibiting RAD52 and/or PARP1 in the tumor cells that are deficient in the canonical repair pathways has been the potential target for inducing the effect of synthetic lethality. Unfortunately, the development of resistance to commonly used PARP1 inhibitors (PARPi) represents the greatest obstacle to working out a successful treatment protocol. DNA polymerase theta (Polθ), encoded by the POLQ gene, plays a key role in an alternative DSB repair pathway—theta-mediated end joining (TMEJ). Thus, it is a promising target in the treatment of tumors harboring deficiencies in homologous recombination repair (HRR), where its inhibition can induce SL. In this review, the authors discuss the current state of knowledge on Polθ as a potential target for synthetic lethality-based anticancer therapies.

## 1. Introduction

The phenomenon of synthetic lethality (SL) was discovered and described for the first time in *Drosophila melanogaster* almost one hundred years ago [1]. One can define synthetic lethality as follows: when pathway A is defective, a redundant pathway B enables cell viability. If pathway B is inactivated or inhibited in cells deficient for pathway A, then both A and B are not functional, which leads to cell death. After 85 years, it was first applied in targeted cancer therapies [2,3]. However, it took time to change it into an efficient treatment protocol. Nowadays, there are several commercialized drugs that utilize this mechanism, which have been approved by the Food and Drug Administration (FDA), for example, olaparib, rucaparib, niraparib, and talazoparib [4]. All of these are poly (ADP-ribose) polymerase 1 inhibitors (PARPi) and their application has successfully been translated into therapy, mainly in combination with homologous recombination deficient (HRD) tumors, not only including BRCA mutations, but also the ones imitating BRCA-mutated cancers, called BRCA-associated or BRCAness. Interestingly, there have also been several cases found of patients with ovarian cancer who did not present these mutations but relapsed platinum-sensitivity disease after the administration of PARPi. However, there are some drawbacks of this solution, namely, cancer cells can develop resistance to PARPi due to homologous recombination repair pathway restoration [4]. Therefore, scientists are continuing their research to explore new synthetic lethal relationships, which can be used to treat drug resistant cancer.

Recent studies demonstrate that DNA polymerase theta (Polθ), encoded by the *POLQ* gene, might play a significant role in alternative DNA double-strand breaks (DSBs) repair pathways [5]. Thus, Polθ is suggested to maintain genome stability, however, its activity is correlated with cancer progression [6]. Accordingly, cancer cells have elevated expression of Polθ, which promotes their survival. Normal cells, however, have expressed a low or a non-existent level of Polθ. Furthermore, silencing Pol in HR-deficient cells reveals a synthetically lethal correlation between Polθ and HR genes. In addition, Polθ depletion causes tumor cells to become more sensitive to other treatments such as radiation or chemotherapy. Polθ is fated to become a new target in customized cancer treatment due to this Polθ feature and its probable engagement in PARPi resistance mechanisms in tumors [7].

In this review, the authors described Polθ and its role in the DSB repair mechanisms as well as focus on the aspect of synthetic lethality in the context of anticancer therapies. Finally, the authors emphasize the promising role of Polθ as a target of this kind of treatment.

Google Scholar and PubMed were used to review the most relevant papers (published until March 2022) that focused on the role of polymerase theta in the context of synthetic lethality and potential anticancer therapy. The authors considered studies performed on animals as well as human subjects (in vivo, in vitro) along with the clinical trials. Keywords applied were as follows: synthetic lethality, dual synthetic lethality, microhomology-mediated end joining, DNA damage response, helicase, polymerase, DNA repair, polymerase theta, cancer, targeted cancer therapy, polymerase theta-mediated end joining, TMEJ, double strand break repair, MMEJ, homologous recombination repair, HR, and anticancer therapy.

## 2. Polθ: Structure and Functions

Mammalian cells are known to contain at least 16 different DNA polymerases that function in semiconservative DNA replication (pols α, δ, ε), base excision repair (pol β), mitochondrial DNA replication, repair and degradation (pol γ), DSB repair and immunological diversity (pols λ, μ, pol θ, and terminal deoxynucleotidyl transferase), and DNA damage tolerance by translesion synthesis [8]. Among them, DNA polymerase theta has the most unique structure. This protein with a mass of 290 kDa is comprised of an N-terminal helicase-like domain (superfamily 2 helicase domain, Polθ-Hel) and a C-terminal DNA polymerase domain (family-A polymerase domain, Polθ-Pol) separated by a long central domain [9]. As such, it is the only eukaryotic DNA polymerase containing a helicase domain [10].

Polθ-Hel shares a close homology to HELQ (helicase Q, POLQ-like helicase, HEL308) belonging to superfamily 2 helicases conserved in eukaryotes and archaea. It is thought to function in the early stages of recombination following replication fork arrest and has a specificity for the removal of the lagging strand in model replication forks [11]. The finding that Polθ has a role in the regulation of DNA replication timing in human cells is noteworthy because this program’s regulation is exceedingly complex. Up to this point, only the loss of Rap-interacting-factor-1 (Rif1) has been found to impact the replication timing of a subset of domains in vertebrates. Rif1 was first identified as a telomere-binding protein in yeast and a double-strand break response factor in mammals, and it has also been proven to regulate replication timing in yeast [12].

HELQ can displace streptavidin from a biotinylated DNA molecule, suggesting that one function of the enzyme may be in the removal of bound proteins at stalled replication forks and recombination intermediates. Similarly, Polθ-Hel has the ability to promote annealing of complementary ssDNA in an ATP-independent manner and RPA-coated ssDNA in an ATP-dependent manner. The latter counteracts RPA activity, which results in the promotion of polymerase theta-mediated end joining (TMEJ). Another similarity between Polθ-Hel and HELQ is the ability to unwind lagging threads on the replication fork in an ATP-dependent manner in 3′→5′ direction. Furthermore, its activity allows short DNA fragments to be disentangled in an ATP-dependent manner in 3′→5′ direction, not unlike DNA HELQ. In TMEJ, the activity of the Polθ-Hel domain drastically increases the yield of the long ssDNA fragments conducted by Polθ-Pol through the inhibition of non-productive snap-back replication [1,13,14].

Polθ-Pol is arranged structurally like most other A-family polymerases, meaning that it contains palm, thumb, and fingers subdomains, which structurally resemble a closed right hand. The domain facilitates DNA synapse formation between the 3′ ssDNA overhangs, then, through the use of the opposing overhang as a template in trans, the protein promotes microhomology-mediated annealing and subsequent extension of the 3′ ssDNA terminus. At the ssDNA/dsDNA junction, the presence of a 5′-terminal phosphate increases the rate of the ssDNA extension step, which indicates that Polθ-polymerase exhibits an affinity to this kind of structure, similar to NHEJ polymerases (i.e., Polμ, Polλ). Finally, another Polθ can perform two gap-filling steps by extending the second overhang [15].

Because most of the secondary structural motifs in the central domain of human Polθ are mostly absent, it is generally considered disordered. However there have been suggestions that it contains two binding sites for RAD51, which is an essential recombinase in HR. According to the same studies, Polθ-Hel also contains a putative RAD51 binding site and may act as an anti-recombinase and suppress HR in favor of TMEJ [11].

## 3. The Role of Polθ in Normal Cells

### 3.1. Mechanisms of DNA Double Strand Break Repair: Where Is Polθ?

DSBs might arise directly due to an exposition to ionizing radiation or indirectly due to ultraviolet radiation, reactive oxygen species, or genotoxic stress (e.g., chemotherapy). This kind of lesion is most cytotoxic, because if left unrepaired, it may lead to transcription and replication blockage. As a result, it can trigger apoptosis or necrosis, genetic rearrangements (mutations, deletions, insertions, translocations), and cause disruption in the course of meiosis, weakened functioning of the immune system, abnormal development of the nervous system, or the development of genetic diseases and cancer [16]. It has been suggested that the loss of the function mutations of genes involved in DSB repair and the activation of a back-up pathway is a source of the therapy-refractory character of some cancer cells.

The choice of the method of DSB repair depends on many factors (i.e., the expression, activity, and availability of repair complex components (regulated, among others, by post-translation modifications such as phosphorylation or poly-ubiquitination)) as well as template availability. In canonical pathways, HR (or simply homologous recombination—HR) can only occur in the G2 and late S phases when a sister chromatid is available, whereas non-homologous end joining—NHEJ (also referred as canonical or classical NHEJ—cNHEJ, or DNA-PK-depended NHEJ—D-NHEJ) is used to repair DSBs during the G1 and early S cell cycle phases [17]. It is estimated that more than 90% of DNA double-strand breaks in mammals are repaired by NHEJ, while most of the damage in yeast and bacteria is repaired by HR [18]. A DNA double-strand break (DSB) repair (DSBR) pathway that employs homologous repeats flanking a DSB is known as single-strand annealing (SSA).

Another DSB repair pathway has been named theta-mediated end joining (TMEJ) because it requires DNA polymerase theta [19]. Apart from this, alternative pathways can be distinguished for both NHEJ and HR: alt-NHEJ, also referred to as microhomology-mediated end joining (MMEJ) and transcription-associated homologous recombination (TA-HR), respectively (Figure 1) [16].

#### 3.1.1. Homologous Recombination Repair (HRR)

Homologous recombination is deemed as the only extremely accurate DSB repair process and is critical during DNA replication. HR has high accuracy because of its use of a homologous DNA template (i.e., sister chromatid). This allows the lost sequence information to be copied with high-fidelity during DNA repair synthesis [20]. Accordingly, HR is specifically active in the S and G2 cell cycle phases when the sister chromatid is available, preserving genome integrity by preventing mutations that can occur if an error-prone pathway is used. Apart from this, homologous recombination is activated during meiosis, in an event called crossing-over, when it switches the sequence information between homologous chromosomes to promote genetic variation in gametes. Thus, major defects in this pathway cause embryonic lethality [21].

The initial events in DSB repair are crucial in deciding which cellular pathway to choose. Ku proteins, for example, have a high affinity for DSBs in non-homologous end-joining (NHEJ), but the MRN complex (MRE11-RAD50-NBS1), in particular, MRE11, competes with the Ku proteins. MRN is a key DNA damage response (DDR) factor that binds to DSB ends, activates the ATM (ataxia telangiectasia mutated) signaling kinase responsible for cell cycle checkpoint regulation and phosphorylation of H2AX histone, and is directly involved in the two primary DSB repair process, HR and NHEJ. Nevertheless, P53 binding protein 1 (51BP1) is one of the first proteins that binds to the ends of DSBs. However, during S/G2 cell cycles, a critical HR factor, BRCA1, suppresses 53BP1 by excluding it from the vicinity of DSBs, thus promoting DSB resection. 53BP1 has the ability to inhibit end resection via effector proteins. Shieldin, a 53BP1 effector complex containing C20orf196 (otherwise known as SHLD1), FAM35A (SHLD2), CTC-534A2.2 (SHLD3), and REV7, has been identified [22]. Shieldin is dependent on 53BP1 and RIF1 to localize to double-strand break sites, and its SHLD2 subunit binds to single-stranded DNA via OB-fold domains that are comparable to those of RPA1 and POT1 [23,24,25]. MRN works in conjunction with CtIP to initiate DNA resection of DSBs by promoting nicks near DSB ends. MRN-CtIP can then excise one of the strands in a 3′-5′ direction toward the break. Then, nucleases such as Dna2 and Exonuclease I, which are more processive, are recruited along with DNA helicases (i.e., Bloom’s helicase) to execute an extended 5′-3′ resection step that ends in long 3′ ssDNA overhangs. These structures are bonded and stabilized by replication protein A (RPA), which results in the activation of ATR (ATM- and Rad3-related) kinase. The activation is mediated by ATRIP (ATR-interacting protein) and plays a crucial role in DDR. BRCA1 is hypothesized to help recruit BRCA2 by forming a complex with PALB2 (i.e., BRCA1-PALB2-BRCA2), which, along with other recombination mediator proteins, mediates the replacement of RPA with RAD51 on ssDNA. RAD51 is a Walker A/B containing recombinase that promotes homology search and strand invasion into the sister chromatid, which serves as a template for DNA repair synthesis catalyzed by polymerases [26]. Another essential cofactor RAD54 regulates RAD51 activity by stabilizing the RAD51 nucleofilament and remodeling of nucleosomes. It also stimulates the homology search and strand invasion activity of RAD51 [27].

#### 3.1.2. Single Strand Annealing (SSA)

Single-strand annealing (SSA) occurs during the S and G2 cell cycle phases and uses DNA homology regions of 8–20 base pairs (bp) in length to connect the ends of widely resected DNA. Overall, SSA is considered to be highly error-prone because it eliminates the DNA fragments between repeats because it eliminates DNA fragments between the repeats as well as one repeat [16,28]. Moreover, it typically occurs between repetitive DNA sequences, and as a result, induces massive deletions and DNA rearrangements that can trigger tumorigenesis. The first steps of SSA and HR are a shared MRN complex with CtIP initiating 5′ → 3′ DNA resection. Other factors such as Exonuclease I and BLM helicase as well as Dna2 helicase/nuclease promote further resection. RPA binds to single-stranded DNA (ssDNA), prevents secondary ssDNA structures, and interacts with a variety of replication and repair factors. RAD52 anneals the homologous sequences flanking the break sites, generating a synapsed intermediate that is subsequently subjected to endonucleolytic cleavage of the 3′ ssDNA tails by ERCC1, forming a complex with XPF [29].

After this, the mechanisms of HR and SSA start to diverge. Particularly, SSA is a RAD51 independent pathway, where the ssDNA overhang homology search and annealing is mediated by RAD52 (RAD52 Double Strand Break Repair Protein). Following annealing, the ERCC1 (Excision Repair Cross Complementation Group 1) interacts with XPF (Xeroderma Pigmentosum, Complementation Group F), creating a complex that removes the flanking unannealed non-complementary 3′ssDNA. Finally, DNA Ligase I (LIG1) ligates the paired DNA ends [27,30].

#### 3.1.3. Nonhomologous End Joining (NHEJ)

In mammalian cells, NHEJ is a significant DSB repair pathway, particularly during the G0/G1 phases of the cell cycle. As previously mentioned, BRCA1 suppresses NHEJ in S and G2 despite the fact that this pathway is functional throughout all of the cell cycle phases except mitosis. Although this pathway is considered to be error-prone because it can cause minor insertions and/or deletions, along with chromosomal translocations, it is overall thought to be important for preventing genome instability [31,32].

NHEJ is initiated by the Ku70/Ku80 heterodimer. Particularly, Ku70/Ku80 binds to the ends of DSBs with high affinity, protecting them from resection, which causes HR. Only after this will the recruitment of all downstream NHEJ factors such as DNA-PKcs (DNA dependent Protein Kinase catalytic subunit), XRCC4 (X-ray repair cross-complementing protein 4)-LIG4(DNA ligase IV)-XLF (XRCC4-Like Factor) complex, and X-family DNA polymerases (Pols) lambda (Pol λ) and mu (Pol μ), be possible [33]. Ku70/80 specifically activates DNA-PKcs, causing it to autophosphorylate and phosphorylate other proteins. During NHEJ, Polλ and Polμ enable DNA extension and gap filling in NHEJ-mediated DNA synapses [34]. XRCC4 is an essential auxiliary factor for LIG4 that boosts its activity responsible for ligation of the ends of DSBs, it is involved in the latter steps of NHEJ. This step is also stimulated by XLF [35].

#### 3.1.4. Polymerase Theta-Mediated End Joining (TMEJ)

Polymerase theta is mainly found to be present in TMEJ, which in itself is an alternative pathway of NHEJ, sometimes treated as a separate, third pathway. The role of Polθ ortholog, Mus308, in the repair of DSB was first discovered in studies of Drosophila. Mus308 was found to play a role in a synthesis-dependent and Ku70/LIG4-independent end joining pathway in vivo break joining assays of DSBs produced by I-SceI homing endonuclease. There are similarities between TMEJ and SSA and HR as they engage a common intermediate, the 3′ssDNA tails generated after the resection of chromosome breaks [36].

The common steps include the activation of helicases such as the MRN complex as phosphorylated CtIP generates 3′ DNA overhangs. Polθ binds this long single strand DNA overhang and anneals to the microhomologous sequences, at least 2 base pairs in length, in order to utilize them as primers for DNA synthesis [6,29]. Polymerization of DNA stabilizes the ends that undergo ligation by either the LIG3-XRCC1 complex or LIG1 alone [37,38,39].

Polθ uses the ssDNA overhang on the opposite side of the break from its binding site as a template for DNA extension, which ultimately results in the stabilization of the intermediate DNA structure formed during the initial phases of the repair process. Subsequently, polymerase is thought to lengthen the second overhang, thereby filling the gap caused by the occurrence of DSB. LIG3 is required for the final fusion of the gap after the processing of the transition structure by other enzymes such as endonuclease [38,39]. Furthermore, due to the lack of proper stabilization, the overhangs elongated by Polθ may detach from the structure. If detached overhangs are then annealed and used as templates once more, the templated insert fragments (additional inserts) are created between the deletion junctions [21].

Polθ has been shown to be able to promote DNA synapse formation from DSB-adjacent 3′ ssDNA containing microhomologous sequences of at least 2 bp in length. In this situation, Pol synthesis is compatible with the extremely efficient capacity of the polymerase to elongate DNA strands from mismatched termini as well as a tendency for primer-template slippage [15].

A different recently published study discovered that Pol-pol DNA endonuclease activity, which has been associated with end-trimming during TMEJ. Furthermore, the polymerase domain has been found to exhibit 5′-deoxyribose phosphate lyase activity, which suggests that it might take part in base excision repair (BER) [40,41,42].

However, under certain circumstances, neither NHEJ nor HR are sufficient, and only TMEJ is able to mend the lesion. For example, the selection of the repair mechanism could be based on DNA end resection. TMEJ is favored in situations when the overhangs of DNA are 30–70 nucleotides long. Additionally, as could be expected, TMEJ is engaged in DNA repair in different HR- or NHEJ-deficient backgrounds such as BRCA1/2 or Ku70 deficiency. This is reflected in various studies showing synthetic lethality between Polθ, the main protein of TMEJ repair, and the canonical DSB repair pathway genes. Despite frequently being described as an alternative NHEJ, TMEJ differs from it substantially as it does not require the presence of Ku heterodimers and is able to act only on resected DNA ends. Polθ utilizes homologous fragments from both sides of the lesion during repair initiation, which invariably results in the loss of one of these fragments. Consequently, TMEJ is an intrinsically mutagenic repair pathway [1,43].

## 4. The Role of Polθ in Malignant Cells

### 4.1. Expression of Polθ in Cancer Cells

The Polθ expression in human malignancies was investigated since it has been linked to chromosomal instability in *D. melanogaster*. Many human neoplasms including those of the lung, stomach, small intestine, rectum, and colon overexpress this protein, which correlates with poor survival [44,45,46,47,48]. It is estimated that approximately 70% of breast cancers are characterized by the overexpression of Polθ [45]. In addition, the expression level of the polymerase is particularly high in lung, breast, and ovarian HR-deficient cancer cells. Due to the fact that Polθ has low or no expression in normal tissue, one can speculate that it constitutes an ideal tumor-specific radiosensitization target. Accordingly, its depletion causes the sensitization of cells deficient in homologous repair to radiation and their decreased viability [7,46,49,50]. Moreover, the knockdown of Polθ with short interfering RNA (siRNA) has been associated with increased DSB formation, destabilization of replication forks, and enhanced sensitivity to some genotoxic agents, suggesting its role as a guardian of the genome [51]. It has been proposed that while keeping a correct level of Polθ expression is crucial for maintaining genome stability, it is possible that the elevated level may favor the occurrence of chromosomal rearrangements, which may ultimately lead to the generation of more aggressive neoplastic phenotypes [52].

Accordingly, the Japanese patients showed not only a significant increase in Polθ expression in colorectal and lung tumors compared to the adjacent tissues, but also an elevated level of Polθ significantly lowered the survival rate within a timeframe of 24 months. Another study of patients with colorectal cancer, but conducted in France, explored the expression levels of 47 genes involved in replication: 18 were downregulated while 17 were upregulated. Not only was *POLQ* among the upregulated genes, but its mRNA level also correlated with a substantial drop in the patients’ survival [6,49]. Based on an analysis of early-stage non-small cell lung tumors, increased Polθ expression within a five-gene prognostic panel was associated with a worse prognosis for patients. Moreover, data from The Cancer Genome Atlas (TCGA) showed that the level of Polθ expression in ovarian cancer was correlated with the degree of tumor development [49,53,54]. Overall, the above data suggest that the presence of the overexpression of Polθ promotes the survival and/or growth of cancer cells.

Still, it is possible that Polθ takes part in yet unknown mechanistic interactions within cancer development, other than DNA repair or replication [45]. An excess of Polθ may deplete cofactors needed for translesion replication or DNA repair pathways such as TMEJ or BER, in which the polymerase is also involved. Decreased resistance to alkylating agents and more frequent short-fragment DNA replication in Polθ overexpressing cells may—over time—lead to DNA damage accumulation and, eventually, to a severe reduction in the efficiency of replication fork progression. Polθ overexpression has also been shown to lead to other neoplastic cell traits such as chromosome end fusions and dicentric chromosomes. This may point toward defects in telomere formation and suggests that Polθ overexpression is a factor for a more aggressive tumor phenotype and a higher likelihood of disease recurrence [23,45,53]. Such traits make Polθ a highly attractive target of clinical interventions [6,42].

### 4.2. HR-Deficient Tumors

It has been discovered by previous research that HR-deficient tumors have a high sensitivity to Polθ, implying that TMEJ is required for HR-deficient tumor survival. To test whether there is a synthetic lethality between the HR genes and POLQ, Ceccaldi et al. created a HR-deficient ovarian cancer cell line. To determine the cell survival, this cell line was depleted of Polθ and exposed to cytotoxic agents. Following exposure to a variety of inhibitors, Polθ depletion reduced the lifespan of these HR-deficient cells, further supporting that HR-deficient cells are dependent on Polθ for survival. Furthermore, in mice, the deletion of HR and Polθ resulted in embryonic lethality [51]. HR-deficiency and high levels of replication stress characterize BRCA1/2-deficient breast and ovarian cancers, leading to genomic instability [55]. BRCA1/2 and Fanconi Anemia (FA) proteins work together in healthy cells to keep replication forks stable and maintain genomic integrity [56]. BRCA1/2-deficient tumors have previously been demonstrated to upregulate FANCD2, a protein necessary for the protection of replication forks, thus protecting DNA strands from excessive nucleolytic degradation [53,54].

In a study conducted by Kais et al., tumors with BRCA1/2 deficiencies were found to exhibit a compensatory increase in the expression of FANC [57]. The function of FANCD2 is to stabilize stalled replication forks and promote alt-EJ repair in tumors deficient in BRCA1/2. FANCD2 knockout in those tumor cells resulted in acute DNA repair defects and an upsurge in cell death. Unstable replication forks cause copy number variation mutations and chromosomal translocations in cells deficient in BRCA1/2 [51]. Genomic instability, while crucial to tumor progression, can also limit cancer cell survival if it is excessive. Thus, mechanisms have evolved in BRCA1/2-deficient cells, which allow them to tolerate genomic instability and replicative stress, ultimately allowing the cells to survive and replicate DNA. For example, in BRCA1/2-deficient cells, an upregulation of the error-prone Polθ/PARP1-mediated alt-EJ DNA repair pathway was found as a mechanism compensation for HR defects [1,2]. According to one study, FANCD2 promotes Polθ recruitment at the sites of damage and TMEJ repair. The loss of FANCD2 in BRCA1/2-deficient tumors increased cell death. This finding shows that FANCD2 and BRCA1/2 have synthetic lethality [58].

### 4.3. Synthetic Lethality Targeting Polθ

Tumor cells that are reliant on alternative backup pathways of DNA repair become novel targets for anticancer treatment based on the phenomenon of synthetic lethality, which is one of the innovative approaches in eliminating cancer cells [45]. In the case of synthetic lethality, losing a gene in a cell involved in a metabolic process that is important for cell survival is compensated for by the action of another gene engaged in a pathway alternative to this process (Figure 2) [59].

The results of studies on primary glioblastoma, melanoma, and other cancer cells revealed that the inhibition of certain DNA repair mechanisms can lead to synthetically lethal effects in cancer cells with simultaneous sparing of normal cells [60,61]. The initial success associated with the use of PARPi in the treatment of ovarian and breast cancer with BRCA1/2 mutations has established a foundation for targeting DSB repair pathways by inducing synthetic lethality [62].

The TMEJ pathway is yet to be defined exactly, and numerous research groups are in the process of seeking therapeutics that may serve to limit Polθ activity, and to utilize this pathway in cancer therapy. In recent research, inhibiting Pol was found to sensitize cells to replication stress produced by drugs such as topoisomerase poisons or ATR inhibitors, leading to new cancer treatment candidates. DSB repair-proficient cancer cells expressing high levels of Polθ may be sensitive to the inhibition of Polθ in combination with standard cytotoxic drugs [8].

Since the DNA synthesis activity of Polθ is essential for TMEJ and the proliferation of DSB repair-deficient cells, it is anticipated that pharmacological inactivation of the Polθ polymerase domain will eliminate HR/NHEJ-deficient cancer cells [63]. Additionally, DSB repair-proficient cancer cells expressing high levels of Polθ may be sensitive to the inhibition of Polθ in combination with standard cytotoxic drugs. The combination of Polθ inactivation, together with PARPi or RAD52i, will exert a synergistic “dual synthetic lethality” in NHEJ- or HR-deficient solid tumors.

It has been proven that Polθ inhibition results in HR repair enhancement and an increase in RAD51 clusters. This can be explained by the fact that Polθ can bind directly to RAD51 causing HR inhibition [51,63]. These findings suggest that Polθ prevents the formation of HR repair complexes, which are toxic in BRCA1/2 defective cells. Both the depletion of Polθ in ovarian cancer cells with inactive FANCD2, a gene crucial for the DNA repair pathway through HR, and the inactivation of these genes in the cells of embryos from mice produced a synthetically lethal effect [64,65]. Moreover, the *POLQ* gene knockout in xenografts prepared from the cells of the tumor with dysfunctional HR genes increased the degree of its sensitivity to PARP inhibitors and enhanced survival in the knockout mice compared to the control group maintaining Polθ expression [66].

## 5. Polθ—A Therapeutic Target for Cancer Therapy

### 5.1. Why Target Polθ to Trigger Synthetic Lethality?

DSB-repair-deficiency depends on Polθ mediated-TMEJ. It is anticipated that the pharmacological inhibition of Polθ will selectively kill cancer cells, which depend on Polθ mediated-TMEJ. Moreover, recent studies have suggested that secondary mutations restoring BRCA1/2 function are caused by the activity of Polθ mediated-TMEJ and that Polθ inhibition can prevent the development of PARPi resistance. PARP1 and RAD52 inhibitors are involved in the already known synthetic lethal mechanism in cancer treatment. Initially, their success was undoubted; however, with the passing of time, research brought about evidence of cells becoming resistant to PARPi [67,68,69]. This has motivated scientists to search for further potential inhibitor targets, and they focused on Polθ. Despite being an important player in DNA DSBR, it has been found that Polθ may be one of the factors responsible for its resistance to radiation and chemotherapeutics including PARP inhibitors (PARPi) [42,51,70,71,72]. Nevertheless, the exact mechanism has not yet been discovered; there are certain assumptions that the engagement of Polθ inhibitors may improve treatment effectiveness.

In the authors’ opinion, the main interest should be focused on the type of tumor with alterations to the HR genes since the studies showed that cells of this type are ‘hyper-dependent’ on Polθ mediated repair (like with PARP1). Moreover, they express higher levels of Polθ, which is attributed to their high survival rate [68,73]. Therefore, the use of Polθi would be a great opportunity to selectively kill these cancer cells. Furthermore, referring to other studies that have shown a broad range of synthetic lethal genes with Polθ, it is also possible to work on other therapeutic strategies than those only known with PARPi and RAD52i [67].

It may look like Polθ acts similarly to PARP1, however, there is still evidence that Polθ inhibitors lead to SL in a different way from PARPi [72]. Additionally, as above-mentioned, HR-deficient cells with a reduced level of Polθ expression are more sensitive to radiation and probably other antitumor agents such as chemotherapy, cisplatin, and mitomycin C. Furthermore, it is possible that several DNA lesions are repaired only via a Polθ mediated mechanism, hence we can hypothesize that cancer cells with such lesions may be even more sensitive during treatment with Polθi [74].

Considering its complex characterization, Polθ appears to be a promising therapeutic target, especially in the case of PARPi resistance. However, this does not mean that Polθ has to replace the PARP and RAD52 inhibitors; rather, it could work simultaneously to enhance their action or serve as the inhibitor’s target itself [75].

### 5.2. Polθ Inhibition as a Potential Target for Synthetic Lethality-Based Anticancer Therapy

Based on the knowledge gained during the research, it has been proposed that the inhibition of Polθ in HR-deficient cells might induce SL and cause the elimination of cancer cell [76]. Moreover, “dual pathway synthetic lethality” expands the synthetic lethal approach to simultaneous targeting of two repair mechanisms. The inhibition of Polθ and PARP1 may lead to a dual SL effect because these types of synthetic lethality interactions involve two or more genes and two pathways [77].

A combination of Polθ inactivation together with PARPi or RAD5i will exert synergistic dual synthetic lethality in c-NHEJ or BRCA1/2 HR-deficient solid tumors. However, not only PARP1, but also other DSB repair proteins (e.g., BRCA1/2, RAD52, and ATM) have a potential synthetic lethal connection to Polθ [76,78]. In total, 140 of these genes have been revealed using a CRISPR genetic screen, which provides scientists with a broad field to search for suitable targets in anticancer therapies [64]. Nevertheless, the mechanisms that underlie these relationships still require more profound research and could indicate a good direction for further studies on Polθ inhibitors (Polθi). Moreover, DNA repair proteins interact differently with Polθ, depending on whether they do so with its helicase or polymerase domain, or can interact with both domains; thus it is important to take into account its structure when considering the Polθi design [19,79,80,81].

The DNA damage repair deficiency has recently been associated with anti-tumor immunity activation, with compelling evidence. BRCA2 inactivation has been shown to induce an innate immune response. Lian Li et al. looked into the link between inactivating POLQ and/or FANCD2, two key DNA damage repair genes, and probable innate immune response activation. In comparison to single POLQ or FANCD2 KOs, double KOs of POLQ and FANCD2, which promote POLQ recruitment at sites of injury, drastically reduced cell proliferation in vitro and in vivo. The POLQ and/or FANCD2 KO esophageal squamous cell carcinoma (ESCC) cells had a considerably higher number of micronuclei. The activation of cGAS (Cycling GMP-AMP Synthase) and the overexpression of interferon-stimulated genes (ISGs) were also seen when POLQ and/or FANCD2 were lost (ISGs). Taken together, these results indicate the potential for the activation of the innate immune response through the cGAS-STING-STAT1 pathway, after the loss of both the Polθ and FANCD2 proteins [6].

Using Polθi may be a promising approach in the treatment of cancer types that have developed resistance to PARP inhibitors due to genetic alterations. This type of inhibition is unlikely to elicit a response in the case of secondary mutations that reactivate BRCA and lead to the recovery of HR activity. However, most relapses of neoplastic diseases result from other changes including decreased expression of DNA repair proteins [82].

In the case of such changes leading to PARPi resistance, the use of Polθ inhibitors could lead to the elimination of tumor cells. Furthermore, the simultaneous inhibition of PARP and Polθ may also provide promising effects. Compared to the use of each of the inhibitors separately, the application of dual inhibition will make it difficult for the tumor to develop resistance. Data from the in vitro studies support this assumption, namely, the use of PARPi in combination with decreased Polθ expression leads to a synergistic reduction in colony formation by cells with inactive BRCA1 [82].

Aside from the synthetically lethal effect, the disruption of Polθ expression also sensitizes cancer cells with dysfunctional HR to the effects of radiotherapy to an extent that may bring significant therapeutic effects. A drastic slowdown in tumor mass growth observed during studies on the inhibition of Polθ and PARP in xenografts also indicates such a synergistic effect [83].

#### Novobiocin and ART558

Novobiocin is a coumarin antibiotic that has lately been identified as a Polθ inhibitor in a small-molecule screen. Novobiocin has already been used in cancer studies, however, with poor results. Novobiocin is a coumarin antibiotic that was recently discovered to be a Pol inhibitor in a small-molecule screen. Cancer studies have already utilized novobiocin, albeit with unsatisfactory results [84].

The first paper concerning it was published in 2021 and confirmed the successful inhibition of Polθ in human BRCA1/BRCA2-deficient cells in vitro and in vivo. Novobiocin is reported to inhibit MMA and abolish the recruitment of the Polθ protein to the DNA damage sites [64]. It has been revealed in experiments conducted by Zhou et al. that novobiocin specifically targets Polθ, and has a similar impact on cells as genetic methods of Polθ depletion. Furthermore, novobiocin enhances the activity of PARP inhibitors, allowing it to bypass the PARPi resistance mechanisms. Furthermore, cells that achieve PARPi resistance through BRCA2 gene somatic reversion are also resistant to novobiocin.

Zatreanu et al. reported that another Polθ inhibitor called ART558 could also eliminate cancer cells and tumors that have become resistant to PARP inhibitors. Using this Polθi in combination with a PARPi in the patients with cancer characterized by mutations in BRCA genes might prevent resistance from emerging in the first place. ART558 has the ability to inhibit theta-mediated end joining—a major Polθ-mediated DNA repair process—without affecting non-homologous end joining. Furthermore, BRCA1/BRCA2 mutant tumor cells have been found to exhibit DNA damage and be affected by synthetic lethality upon exposure to ART558. ART558 also enhances the PARP inhibitor effects.

Nevertheless, ART558 has great potential, but so far, it has not been used in vivo due to its poor stability in a rat model. Therefore, another Polθi, ART812 was used in these experiments. It has been revealed through genetic perturbation screening that the 53BP1/Shieldin complex has defects, which result in PARP inhibitor resistance, eliciting both in vitro and in vivo sensitivity to small-molecule Polθ inhibitors. The mechanism of ART558 action involves increasing the ssDNA biomarkers and synthetic lethality in cells with 53BP1 defects. The inhibition of DNA nucleases, and thus the inhibition of end resection, is able to reverse these effects, thus implicating them in the mechanism of synthetic lethality.

Importantly, these two inhibitors have different mechanisms of action: novobiocin targets the ATP-ase domain and ART558 the polymerase domain of Polθ, which could be the advantage in advancing the research on Polθi.

Both novobiocin and ART558 represent powerful tools that prove the relevance of targeting Polθ in the case of cancer. It is still unknown whether these drugs will benefit people with cancer, but the strategy of inhibiting Polθ in tumors with defects in homologous recombination appears promising [63].

### 5.3. Future Perspectives

Personalized anticancer therapy can result in increased treatment effectiveness and reduced toxic effects. To achieve this goal, it is necessary to create a therapeutic model based not only on the clinical indicators of a specific neoplastic disease, but also on the molecular biology of cancer. Identifying the elements that a personalized therapy will target in diverse types of cancer is still a serious challenge for both researchers and clinicians. The use of carefully selected inhibitors of DNA double-strand break repair proteins with an intention to induce cell death based on the phenomenon of synthetic lethality is a promising approach, in which personalized medicine will be used to treat human solid tumors.

The initial success achieved with PARP inhibitors such as lynparza has indicated a promising direction of treating some patients with tumors having mutations in BRCA1/2. It has also established evidence that supports the concept of DSB repair by inducing synthetic lethality. However, over time, these cancer types invariably become resistant to drugs. Recent studies have shown that Polθ becomes essential in the cells that are deficient in factors facilitating the canonical DSB repair mechanism (BRCA1, BRCA2, Ku70), which indicates the backup function of Polθ-dependent DNA repair processes. Due to this discovery, more attention is now being paid to Polθ as a new therapeutic target. Currently, new genes involved in the DNA damage repair mechanism, chromatin structure maintenance, and DNA metabolism are recognized as synthetic lethality partners for Polθ [36,81,85]. It is predicted that the pharmacological inhibition of Polθ selectively kills TMEJ-dependent cancer cells mediated by Polθ. Moreover, recent studies suggest that secondary mutations restoring the BRCA1/2 function are caused by the activity of TMEJ with Polθ mediation. In this case, the inhibition of Polθ may prevent the development of resistance to PARPi.

Currently, clinical trials on an anticancer drug from the group of Polθ inhibitors started at the end of 2021.

## Figures and Tables

**Figure 1 genes-13-01101-f001:**
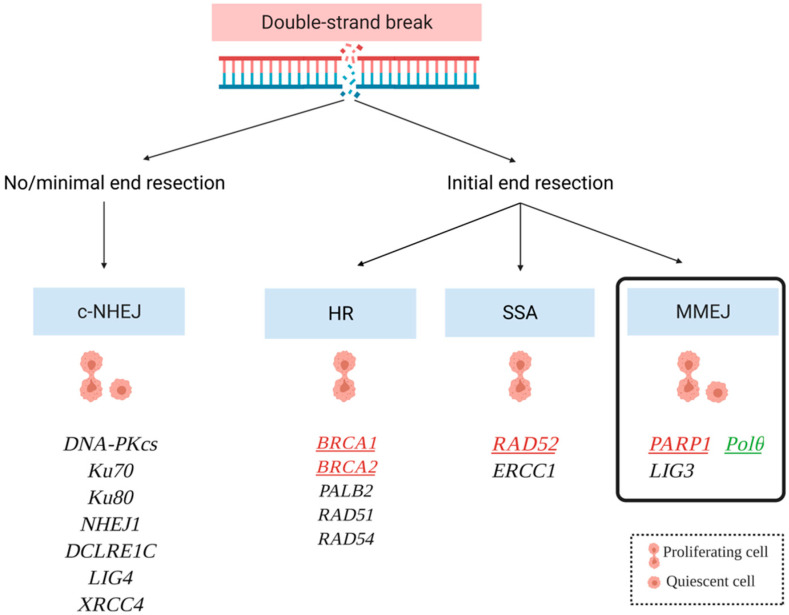
Double-strand break (DSB) repair mechanisms in the quiescent and proliferating cells and major proteins participating in them. Homologous recombination (HR), single-strand annealing (SSA), and microhomology-mediated end joining (MMEJ), in contrast to canonical non-homologous end joining (c-NHEJ), require DNA end resection to expose 3′ single stranded DNA fragments. Polθ as a potential target for synthetic lethality-based therapy has been marked in green. The PARP1, Rad52, BRCA1, BRCA2 partners for dual synthetic lethality have been marked in red.

**Figure 2 genes-13-01101-f002:**
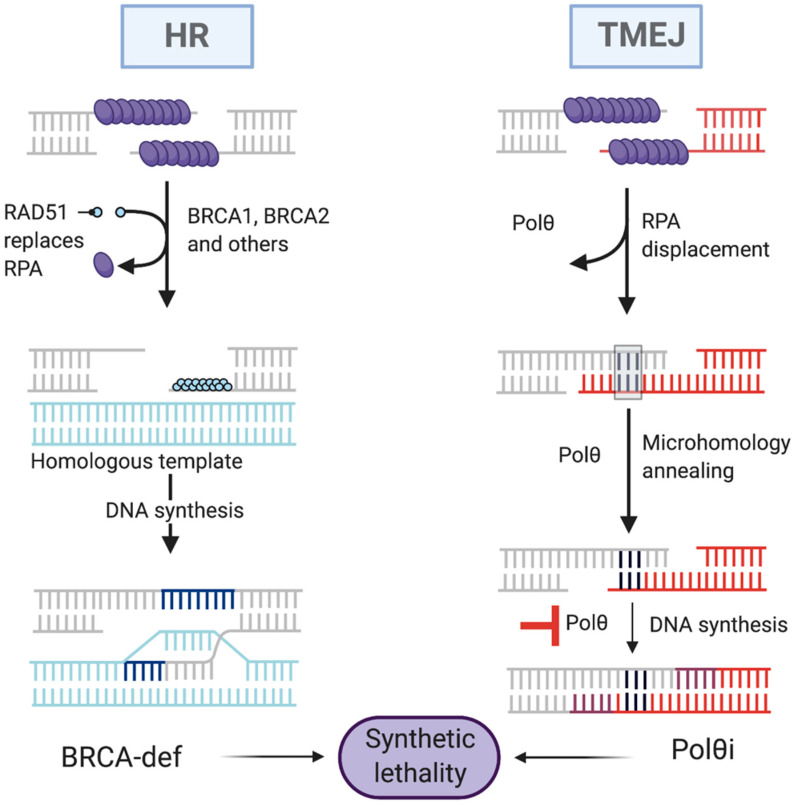
The synthetic lethality (SL) strategy in homologous recombination (HR) deficient cells.

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
