# Peer review of "Synthetic Lethality Targeting Polθ"

_genes, 2022, doi:10.3390/genes13061101_

Round 1
Reviewer 1 Report
Synthetic lethality, initially described in Drosophila, is known as a lethal phenotype, in which single gene mutation/ deficiency is viable, whereas two genes perturbation results in cellular/ organism death. Synthetic lethality is a well-defined concept to study genetic interactions and develop anticancer drug targets. Remarkably, synthetic lethality between different DNA damage repair pathways has been successfully used for cancer therapy. For instance, BRCA1/2 mutated tumors confer sensitivity to PARP inhibitors. However, the cancer cells can develop PARP inhibitor resistance due to various reasons. It is urgent to develop new therapeutic strategies and new drug targets, for example, polymerase theta. It is important to summarize our understanding of the concept of polymerase theta mediated cancer vulnerability (or fitness). Unfortunately, there are obvious flaws, resulting in a poor-quality manuscript. For instance: certain conclusion is not correct; some references/ citation is not correct; some descriptions can be improved, and more…..
- Several references format should be corrected, for example:1, 2, 4, 8, 51, 73, 76, 77, 81
- Line 20: Rad52 might be a potential target for cancer therapy, but definitely not a main target yet.
- Line 51-53: please cite proper research articles.
- Line 55-56: this is not an appropriate conclusion about “why polymerase theta is recognized as a new target”.
- Line 61-69: why the authors include how they search literature??
- Line 75: reference 10-12 is not correct.
- Line 76: fusion protein???
- Line 84-86: the authors should review Pol θ function during replication rather than HELQ. By the way, proper citation is needed.
- Line 112: Do the authors mean DNA double strand break repair?
- Line 116: please refer to proper reference.
- Line 126-130 & Line 201-203: NHEJ occurs throughout interphase, but not in mitosis. Reference #30 should be corrected.
- Line 132-133: the description of SSA is not correct.
- Line 134-137: not quite understand what the authors want to say. Meanwhile, the reference is not correct.
- Figure 1: the major players in different repair pathway should be more representative. And this figure cannot show why Pol θ is a potential target for synthetic lethality-based therapy.
- Sections 3.1.2 and 3.1.4 are poorly organized, the summary of HR and NHEJ is confusion.
- Line 154-155: the cause and effect of “crossing-over” during meiosis is not correct.
- Line 165: regarding to 53BP1 downstream effectors, authors should include the well-defined Shieldin complex.
- Line 186-187: this is not the reason why SSA is error prone. Importantly, flap cleavage, the key step of SSA, is neglected in this section.
- Line 219: why Pol θ is involved in every DSBs repair mechanism?
- Line 231: LIG2-XRCC1, LIG3 (rather than LIG2) could be more commonly used.
- Line 248-251: citation?
- Line 252-258: the authors should better define the situations that TMEJ is more likely to repair the lesions compared to HR and NHEJ.
- Line 262: “Drosophila melanogaster” should be “Drosophila melanogaster”
- Line 264-265: the citation should be (Lemee, et al. PNAS, 2010) and others.
- Line 277-282: the corresponding studies should be cited properly.
- Line 292: the authors didn’t describe the Pol θ role in BER.
- Section 3.2: the authors should summarize that Pol θ overexpression/ mutation contributes to poor clinical outcomes in different studies (large cohort like French, Japanese).
- Line 308: please correct this sentence.
- Line 314-327: the relationship between BRCA1 and FANCD2 - Pol θ should be more precisely summarized. Line 321: key reference missing.
- Line 335-340: the correct references should be appreciated.
- Line 350: please cite research articles.
- Title of section 4.1 is confusing.
- Line 378: DSBR rather than DSB.
- Line 383-386: the authors may also need to consider the published literatures about the Pol θ dependency under different genetic backgrounds.
- Line 404: the authors may need a better explanation of “dual SL”.
- The genetic interaction between Pol θ and FANCD2 is confusing. FANCD2 promotes Pol θ recruitment upon damage; while FANCD2 and Pol θ double knockout shows synergetic effects in terms of innate immune response. The authors need further clarify this argument.
- Novobiocin and ART558 are two Pol θ inhibitors. However, the description of these two inhibitors (studies) are chaotic. And the description of these two inhibitors should be fair, and shouldn’t be tendentious.
Author Response
Synthetic lethality, initially described in Drosophila, is known as a lethal phenotype, in which single gene mutation/ deficiency is viable, whereas two genes perturbation results in cellular/ organism death. Synthetic lethality is a well-defined concept to study genetic interactions and develop anticancer drug targets. Remarkably, synthetic lethality between different DNA damage repair pathways has been successfully used for cancer therapy. For instance, BRCA1/2 mutated tumors confer sensitivity to PARP inhibitors. However, the cancer cells can develop PARP inhibitor resistance due to various reasons. It is urgent to develop new therapeutic strategies and new drug targets, for example, polymerase theta. It is important to summarize our understanding of the concept of polymerase theta mediated cancer vulnerability (or fitness). Unfortunately, there are obvious flaws, resulting in a poor-quality manuscript. For instance: certain conclusion is not correct; some references/ citation is not correct; some descriptions can be improved, and more….
- Several references format should be corrected, for example:1, 2, 4, 8, 51, 73, 76, 77, 81
- Line 20: Rad52 might be a potential target for cancer therapy, but definitely not a main target yet.
- Line 51-53: please cite proper research articles.
- Line 55-56: this is not an appropriate conclusion about “why polymerase theta is recognized as a new target”.
- Line 61-69: why the authors include how they search literature??
- Line 75: reference 10-12 is not correct.
- Line 76: fusion protein???
- Line 84-86: the authors should review Pol θ function during replication rather than HELQ. By the way, proper citation is needed.
- Line 112: Do the authors mean DNA double strand break repair?
- Line 116: please refer to proper reference.
- Line 126-130 & Line 201-203: NHEJ occurs throughout interphase, but not in mitosis. Reference #30 should be corrected.
- Line 132-133: the description of SSA is not correct.
- Line 134-137: not quite understand what the authors want to say. Meanwhile, the reference is not correct.
- Figure 1: the major players in different repair pathway should be more representative. And this figure cannot show why Pol θ is a potential target for synthetic lethality-based therapy.
- Sections 3.1.2 and 3.1.4 are poorly organized, the summary of HR and NHEJ is confusion.
- Line 154-155: the cause and effect of “crossing-over” during meiosis is not correct.
- Line 165: regarding to 53BP1 downstream effectors, authors should include the well-defined Shieldin complex.
- Line 186-187: this is not the reason why SSA is error prone. Importantly, flap cleavage, the key step of SSA, is neglected in this section.
- Line 219: why Pol θ is involved in every DSBs repair mechanism?
- Line 231: LIG2-XRCC1, LIG3 (rather than LIG2) could be more commonly used.
- Line 248-251: citation?
- Line 252-258: the authors should better define the situations that TMEJ is more likely to repair the lesions compared to HR and NHEJ.
- Line 262: “Drosophila melanogaster” should be “Drosophila melanogaster”
- Line 264-265: the citation should be (Lemee, et al. PNAS, 2010) and others.
- Line 277-282: the corresponding studies should be cited properly.
- Line 292: the authors didn’t describe the Pol θ role in BER.
- Section 3.2: the authors should summarize that Pol θ overexpression/ mutation contributes to poor clinical outcomes in different studies (large cohort like French, Japanese).
- Line 308: please correct this sentence.
- Line 314-327: the relationship between BRCA1 and FANCD2 - Pol θ should be more precisely summarized. Line 321: key reference missing.
- Line 335-340: the correct references should be appreciated.
- Line 350: please cite research articles.
- Title of section 4.1 is confusing.
- Line 378: DSBR rather than DSB.
- Line 383-386: the authors may also need to consider the published literatures about the Pol θ dependency under different genetic backgrounds.
- Line 404: the authors may need a better explanation of “dual SL”.
- The genetic interaction between Pol θ and FANCD2 is confusing. FANCD2 promotes Pol θ recruitment upon damage; while FANCD2 and Pol θ double knockout shows synergetic effects in terms of innate immune response. The authors need further clarify this argument.
- Novobiocin and ART558 are two Pol θ inhibitors. However, the description of these two inhibitors (studies) are chaotic. And the description of these two inhibitors should be fair, and shouldn’t be tendentious.
Author Response
Dear Sir or Madam,
Thank you kindly for your considerations. We have implemented all of your suggestions in our revision of the manuscript, to the best of our ability. We sincerely hope that you find the corrections sufficient to bring the manuscript up to a standard adequate for being published in GENES.
Listed below are our answers and revisions implemented in response to each provided comment:
POINT 1: “Several references format should be corrected, for example:1, 2, 4, 8, 51, 73, 76, 77, 81”
RESPONSE 1: Thank you I have corrected the references you have listed.
POINT 2: “Line 20: Rad52 might be a potential target for cancer therapy, but definitely not a main target yet.”
RESPONSE 2: I have clarified that Rad52 is only a potential target for cancer therapy, and not the main target yet.
POINT 3: “Line 51-53: please cite proper research articles”
RESPONSE 3: Added missing in-text citation
POINT 4: “Line 55-56: this is not an appropriate conclusion about “why polymerase theta is recognized as a new target”
RESPONSE 4: We agree with this statement from the review and have modified our inaccuracies.
POINT 5: “Line 61-69: why the authors include how they search literature??”
RESPONSE 5: Thank you for your comment. We have followed the guidelines for a systematic review such as PRISMA. Even within the framework of a narrative review it is important to specify the bibliographical database(s) used, the period during which the research for the writing of the review was carried out.
POINT 6: “Line 75: reference 10-12 is not correct”
RESPONSE 6: Added missing in-text citation
POINT 7: “Line 76: fusion protein???”
RESPONSE 7: Thank you for pointing our error, the mention of fusion protein has been removed from our manuscript.
POINT 8: “Line 84-86: the authors should review Pol θ function during replication rather than HELQ. By the way, proper citation is needed”
RESPONSE 8: Thank you. The proper citation has been added, along with further information about Pol θ function during replication.
POINT 9: “Line 112: Do the authors mean DNA double strand break repair?”
RESPONSE 9: Thank you. The chapter title has been edited to include the word repair.
POINT 10: “Line 116: please refer to proper reference.”
RESPONSE 10: Added missing in-text citation
POINT 11: “Line 126-130 & Line 201-203: NHEJ occurs throughout interphase, but not in mitosis. Reference #30 should be corrected”
RESPONSE 11: Thank you. We have corrected the information relating to NHEJ to reflect that it occurs throughout interphase, and additionally rectified reference #30
POINT 12: “Line 132-133: the description of SSA is not correct.”
RESPONSE 12: Thank you for making us aware of these important mistakes. The misinformation related to description of SSA has been corrected.
POINT 13: “Line 134-137: not quite understand what the authors want to say. Meanwhile, the reference is not correct.”
RESPONSE 13: Thank you for pointing our error. We have removed this sentence.
POINT 14: “Figure 1: the major players in different repair pathway should be more representative. And this figure cannot show why Pol θ is a potential target for synthetic lethality-based therapy.”
RESPONSE 14: Thank you for your comment. Our aim in creating Figure 1 was to point out potential partners of Pol θ for synthetically lethal interactions. The basis for Pol θ being a potential target for synthetic lethality-based therapy is outlined in Figure 2.
POINT 15: “Sections 3.1.2 and 3.1.4 are poorly organized, the summary of HR and NHEJ is confusion”
RESPONSE 15: Thank you for your comment. We agree with this statement and reorganized sections.
POINT 16: “Line 154-155: the cause and effect of “crossing-over” during meiosis is not correct.”
RESPONSE 16: Thank you for your comment. We based the cause and effect outlined in Lines 154-155 on (National Human Genome Research Institute) “Crossing over, as related to genetics and genomics, refers to the exchange of DNA between paired homologous chromosomes (one from each parent) that occurs during the development of egg and sperm cells (meiosis). This process results in new combinations of alleles in the gametes (egg or sperm) formed, which ensures genomic variation in any offspring produced.”
POINT 17: “Line 165: regarding to 53BP1 downstream effectors, authors should include the well-defined Shieldin complex.”
RESPONSE 17: We agree that this information should be added. We have now included the well-defined Shieldin complex.
POINT 18: “Line 186-187: this is not the reason why SSA is error prone. Importantly, flap cleavage, the key step of SSA, is neglected in this section”
RESPONSE 18: Thank you for drawing attention to this oversight, we have corrected this portion to include the key step of SSA.
POINT 19: “Line 219: why Pol θ is involved in every DSBs repair mechanism?”
RESPONSE 19: We have removed the speculation of involvement of Pol θ in every DSBs repair mechanism.
POINT 20: “Line 231: LIG2-XRCC1, LIG3 (rather than LIG2) could be more commonly used”
RESPONSE 20: We have modified the terminology to reflect LIG3 instead of LIG2.
POINT 21: “Line 248-251: citation?”
RESPONSE 21: We agree a citation is needed and have since added the missing in-text citation.
POINT 22: “Line 252-258: the authors should better define the situations that TMEJ is more likely to repair the lesions compared to HR and NHEJ.”
RESPONSE 22: Thank the reviewer for pointing this out and have further defined the situations that TMEJ is more likely to repair the lesions compared to HR and NHEJ.
POINT 23: “Line 262: “Drosophila melanogaster” should be “Drosophila melanogaster”
RESPONSE 23: We have fixed this error and have italicized the text.
POINT 24: “Line 264-265: the citation should be (Lemee, et al. PNAS, 2010) and others”
RESPONSE 24: Thank you we have corrected the references
POINT 25: “Line 277-282: the corresponding studies should be cited properly.”
RESPONSE 25: Thank you we have corrected the references.
POINT 26: “Line 292: the authors didn’t describe the Pol θ role in BER.”
RESPONSE 26: We agree with the reviewer that describing the pol Pol θ role in BER would be helpful. However, after further research, the involvement of PolQ in BER is a matter of debate, and thus we have decided to omit the inclusion.
“Based on the presence of a weak 5′-deoxyribose phosphate lyase activity in its polymerase domain, POLQ was suggested to act in base excision repair (BER), although the extent of its involvement is a matter of debate”
POINT 27: “Section 3.2: the authors should summarize that Pol θ overexpression/ mutation” contributes to poor clinical outcomes in different studies (large cohort like French, Japanese).
RESPONSE 27: Thank the reviewer for pointing this out. We have added summary.
POINT 28: “Line 308: please correct this sentence.”
RESPONSE 28: Thank you we have corrected this sentence.
POINT 29: “Line 314-327: the relationship between BRCA1 and FANCD2 - Pol θ should be more precisely summarized.”
RESPONSE 29: Thank you. We agree with this statement and have precisely summarized the relationship between BRCA1 and FANCD2 - Pol θ
POINT 30: “Line 321: key reference missing.”
RESPONSE 30: Added missing in-text citation.
POINT 31: “Line 335-340: the correct references should be appreciated.”
RESPONSE 31: Added missing in-text citation.
POINT 32: “Line 350: please cite research articles.”
RESPONSE 32: Added missing in-text citation.
POINT 33: “Title of section 4.1 is confusing.”
RESPONSE 33: Thank you. We have modified the title of section.
POINT 34: “Line 378: DSBR rather than DSB.”
RESPONSE 34: We agree with this statement and have modified our inaccuracies.
POINT 35: “Line 404: the authors may need a better explanation of “dual SL””
RESPONSE 35: Thank you. Detailed explanation of dual synthetic lethality has been added to revised manuscript.
POINT 36: “The genetic interaction between Pol θ and FANCD2 is confusing. FANCD2 promotes Pol θ recruitment upon damage; while FANCD2 and Pol θ double knockout shows synergetic effects in terms of innate immune response. The authors need further clarify this argument.”
RESPONSE 36: Thank you for this suggestion. We have modified the text.
POINT 37: “Novobiocin and ART558 are two Pol θ inhibitors. However, the description of these two inhibitors (studies) are chaotic. And the description of these two inhibitors should be fair, and shouldn’t be tendentious.”
RESPONSE 37: We appreciate the reviewer’s suggestion, and we have modified section 5.2.1. Novobiocin and ART558
Thank you for your kind guidance and cooperation.
Sincerely,
Małgorzata Drzewiecka
Reviewer 2 Report
General Comments
Synthetic lethality is typically achieved in HR deficient tumor by targeting PARP1 or RAD52 with specific inhibitors and leads to clinical responses that are highly efficacious. However, almost uniformly, resistance to commonly used PARP1 inhibitors develops, severely compromising treatment outcome. In the present paper the authors review the potential of DNA polymerase theta (Polθ) as a synthetic lethality target in human tumors with defects in broad aspects of HR.
The paper summarizes useful information and should be of interest to readers of the Journal working in the field. Indeed, the paper covers the topic at different levels of inquiry including clinical trials and promising approaches. However, overall the review gives the impression that it was hastily prepared, with inadequate attention to detail. This will need to be improved. While all sections require careful overhaul, the sections focusing on the molecular functions of the protein will require the most attention, as they contain inaccurate statements and errors.
Specific Comments
- There are two subheadings of 3 in the review.
- Line 145: The subheading should start from 3.1.1.
- Line 107-108: Section lacking the relevant references.
- Line 107: In part 3.1, the authors attempt to outline the role of Polθ in DSB repair. There are only a few words that introduce the function of Polθ in the TMEJ pathway. The content of this section does not fit its title. Overall this section and other section containing mechanistic information will require careful re-writing.
- Line 111: In the section "Role of Polθ in normal cells", the authors do not present the expression or other functions of Polθ in normal cells versus tumor cells. Inclusion of this information will be helpful to the reader. Also cell cycle information will be relevant.
- Line 139: Figure 1, why do the authors compare DSB repair mechanisms in quiescent and proliferating cells? Please explain!
- Line 219: The content of the section is not compatible with the statement here.
- Lines 227-231: There are inaccuracies in the statements here. Please check and correct.
- Line 404: The definition of SL from National Cancer Institute is: Describe a situation in which mutations (changes) in two genes together result in cell death, but a mutation in either gene alone does not. Cancer cells that only have one mutated gene in a specific pair of genes can depend on the normal partner gene for survival. Interfering with the function of the normal partner gene may cause cancer cells to die. Why might simultaneous inhibition of Polθ and PARP1 lead to a dual SL effect? Please explain, adapt and unify definitions.
- Line 425: PolQ is the gene. Use of Polθ may be better here.
- Throughout this review the statement, “It has been proven that…”, appears frequently, but often literature is not cited. It would be important to add the source of the stated knowledge to help the reader follow up.
- Overall: Use the same abbreviation for the same term throughout! I find HRR, HR etc. Also, for other abbreviations: Please, make an effort to use them consistently once defined.
- Check references; many are incomplete!
Reviewer 3 Report
The authors present a review of different repair mechanisms with the focus on PolQ.
Specific points
Around line 219, who speculated that PolQ was involved in all pathways? This needs to be removed or reworded.
The sentence structure around line 307 is confusing.
In the last 18 months there have been several reviews published on PolQ and TMEJ. There is nothing incorrect in this review but the harsh truth is there isn't anything that hasn't been covered in the others.
Author Response
The authors present a review of different repair mechanisms with the focus on PolQ.
Specific points
- Around line 219, who speculated that PolQ was involved in all pathways? This needs to be removed or reworded.
- The sentence structure around line 307 is confusing.
- In the last 18 months there have been several reviews published on PolQ and TMEJ. There is nothing incorrect in this review but the harsh truth is there isn't anything that hasn't beencovered in the others.
Author Response
Dear Sir or Madam,
Thank you kindly for your considerations. We have implemented all of your suggestions in our revision of the manuscript, to the best of our ability. We sincerely hope that you find the corrections sufficient to bring the manuscript up to a standard adequate for being published in GENES.
Listed below are our answers and revisions implemented in response to each provided comment:
POINT 1: “Around line 219, who speculated that PolQ was involved in all pathways? This needs to be removed or reworded.”
RESPONSE 1: Thank you. We have removed the speculation of involvement of Pol θ in every DSBs repair mechanism.
POINT 2: “The sentence structure around line 307 is confusing.”
RESPONSE 2: Thank you for pointing out the confusion surrounding line 307, we agree that it should be adjusted, and we have since done so.
POINT 3: “In the last 18 months there have been several reviews published on PolQ and TMEJ. There is nothing incorrect in this review but the harsh truth is there isn't anything that hasn't beencovered in the others. “
RESPONSE 3: Thank you for your insight. You have raised an important point here, however, we believe that our paper goes beyond other reviews published on PolQ and TMEJ. We hold that our paper surpasses the information contained in any singular paper and provides a further exploration of PolQ in cancer therapy. The paper covers the topic at different levels of inquiry including clinical trials and promising approaches We trust this because we included significant discoveries relating to all pathways in which PolQ is involved, the targeting of PolQ in cancer therapy using dual SL, and the two inhibitors that target PolQ.
Thank you for your kind guidance and cooperation.
Sincerely,
Małgorzata Drzewiecka
Round 2
Reviewer 1 Report
The references should be corrected. Please cite the original paper most related to the description rather than review papers. Format of the reference should also be corrected for further consideration.
Author Response
Dear Sir or Madame,
We appreciate the time and effort that you have dedicated to providing your valuable feedback on our manuscript. We are grateful for your insightful comments. We have been able to incorporate changes to reflect the suggestions provided by the reviewer.
Thank you for your kind guidance and cooperation.
Sincerely,
Małgorzata Drzewiecka
Reviewer 2 Report
The authors have amended the manuscript according to the comments of the original review.
One minor issue the authors will need to address is the format of the references. They should edit them for consistency.
Author Response

(The authors gave the same response as above.)
